# Diagnostic Challenges and Perinatal Outcomes: A Case Series on a Retrospective Study

**DOI:** 10.3390/diagnostics15111329

**Published:** 2025-05-26

**Authors:** Carmen Maria Moral-Moral, Lorena Porras-Caballero, Marta Blasco-Alonso, Celia Cuenca-Marín, Susana Monis-Rodriguez, Ernesto Gonzalez-Mesa, Isidoro Narbona-Arias, Jesus S. Jimenez-Lopez

**Affiliations:** 1Obstetrics and Gynecology Department, Hospital Materno-Infantil, Hospital Regional Universitario Málaga, Avenue Arroyo de los Ángeles S/N, 29011 Málaga, Spain; carmenmmoral14@gmail.com (C.M.M.-M.); lorena27796@outlook.com (L.P.-C.); cecuman2@yahoo.es (C.C.-M.); susanamonis@hotmail.com (S.M.-R.); egonzalezmesa@gmail.com (E.G.-M.); dr.narbona@gmail.com (I.N.-A.); jesuss.jimenez@uma.es (J.S.J.-L.); 2Department of Surgical Specialties, University of Malaga, 29010 Málaga, Spain; 3Research Group in Maternal-Fetal Medicine Epigenetics Women’s Diseases and Reproductive Health, Biomedical Research Institute of Malaga (IBIMA), 29071 Málaga, Spain

**Keywords:** succenturiate placenta, velamentous cord insertion, prenatal ultrasound, vasa previa, maternal–fetal medicine, placental anomalies, Doppler sonography

## Abstract

Succenturiate placenta is a rare anatomical variant characterized by one or more accessory lobes connected to the main placental mass by fetal vessels. While frequently asymptomatic, this condition can lead to serious maternal–fetal complications if not diagnosed prenatally. Early detection through advanced ultrasonographic techniques plays a critical role in guiding obstetric management and reducing adverse outcomes. **Objective:** To describe and analyze the prenatal diagnosis, sonographic characteristics, clinical management, and maternal–fetal outcomes of succenturiate placenta cases diagnosed over a ten-year period at a tertiary care center. **Methods:** We conducted a retrospective observational study of nine pregnancies diagnosed with succenturiate placenta between 2014 and 2024. Data collected included maternal demographics, ultrasound findings, type of cord insertion, presence of associated anomalies such as velamentous cord insertion or vasa previa, vaginal or cesarean delivery, complications, and neonatal outcomes. Ultrasound evaluation was scored based on a four-point checklist assessing key diagnostic steps. **Results:** Five of the nine cases (55.6%) presented isolated succenturiate placenta, while four (44.4%) were associated with velamentous cord insertion. No cases of vasa previa were identified. Obstetric outcomes included three vaginal deliveries (33.3%), two instrumental (22.2%), and four cesarean sections (44.4%), one of which was emergent due to fetal distress. Complications occurred in 44.4% of cases, with intrapartum bradycardia being the most common. One neonatal death was reported due to placental abruption. The quality of the ultrasound diagnosis was high in most cases, though transvaginal scanning was inconsistently applied. **Conclusions:** Prenatal identification of succenturiate placenta via detailed ultrasound, including color Doppler and targeted assessment of cord insertion, is essential to minimize risks associated with this condition. Standardized diagnostic protocols can improve detection rates and enable timely clinical decisions, ultimately improving maternal and neonatal outcomes.

## 1. Introduction

Placental development is a dynamic and multifaceted process that begins during the second week of gestation and extends through the third or fourth month of pregnancy. This process originates from specialized group of cells collectively known as the trophoblast, which forms the outermost layer of the blastocyst and plays a crucial role in implantation and subsequent placental morphogenesis.

Once fully developed, the placenta may undergo minor changes throughout the remaining pregnancy, some of which are detectable and assessable via ultrasonographic studies. These changes may include variations in placental size, thickness, degree of maturation, and location [1].

Abnormalities in placental implantation are frequently associated with vaginal bleeding during the second and third trimesters, significantly increasing both maternal and fetal morbidity and mortality. This scenario represents a considerable challenge in prenatal diagnoses, necessitating the implementation of specific protocols for early detection and appropriate management [2].

The term succenturiate is derived from the Latin word succenturio, meaning “substitute” or “replacement”. A succenturiate placenta is defined as a placenta composed of one or more accessory lobes that are physically separated from the main placental body. These lobes may be connected by fetal blood vessels traversing the fetal membranes. As these vessels are only protected by the membranes, they are particularly vulnerable to vascular accidents during pregnancy or delivery [3,4].

Several epidemiological studies have identified several factors associated with an increased incidence of succenturiate placenta. Recent research reports have reported higher prevalence among women with a history of pelvic infections, those who have undergone assisted reproductive techniques, and individuals of advanced maternal age. In the latter group, the underlying mechanism may involve progressive vascular deterioration or focal placental insufficiency, which may lead to compensatory anatomical adaptation [5].

The ultrasonographic diagnosis of succenturiate placenta is based on the identification of accessory lobes with echogenicity similar to that of the main placenta but located separately and typically smaller in size. Particular concern arises when in the accessory lobe is located a low position in the uterus or crosses the internal cervical os, significantly increasing the risk of complications. Color Doppler imaging enables the visualization of fetal vessels connecting the lobes. The umbilical cord is usually inserted in the main placenta, but marginal or velamentous insertions should not be ruled out, as they significantly increase perinatal risk [6].

Regarding prenatal management, current clinical guidelines recommend thorough evaluation of the entire uterine cavity before determining final placental location. Upon identification of an accessory lobe, it is essential to establish the exact site of umbilical cord insertion and perform transvaginal ultrasound with color Doppler to exclude vasa previa and identify submembranous connecting vessels, which are particularly susceptible to injury [7].

The prognosis of succenturiate placenta is generally favorable when detected as an isolated sonographic finding. However, the risk profile changes dramatically when associated with velamentous cord insertion, a combination classified as type 2 vasa previa. Velamentous insertion is a notable feature in a significant proportion of these cases [4].

The most common complications associated with succenturiate placenta include fetal or maternal death due to rupture of accessory vessels during uterine contractions, potentially resulting in severe hemorrhage and placental abruption. Other complications include preterm birth due to emergency cesarean section for acute fetal distress, retained placenta during the third stage of labor, and an increased risk of postpartum hemorrhage due to retained placental fragments [8].

The obstetric approach to succenturiate placenta does not differ significantly from standard management, except in cases involving associated vasa previa. In such cases, current clinical guidelines recommend planned cesarean delivery before the onset of labor, preferably between 34 and 36 weeks of gestation, following corticosteroid administration for fetal lung maturation in selected cases [9,10].

The aim of this study is to deepen the understanding of prenatal ultrasonographic diagnosis of succenturiate placenta, as well as to provide a detailed analysis of the potential complications arising from this condition, with the ultimate goal of establishing appropriate follow-up protocols that minimize the associated maternal and fetal risks inherent to this placental anomaly.

## 2. Materials and Methods

Study Design and Population. We conducted a retrospective observational study of pregnancies diagnosed with succenturiate placenta at the Department of Obstetrics and Gynecology of the Hospital Materno Infantil of Málaga between January 2014 and December 2024. Case identification was performed through the institutional electronic medical records system, yielding a total of 9 cases.

Inclusion and Exclusion Criteria. All pregnancies with a confirmed prenatal ultrasound diagnosis of succenturiate placenta managed at our institution during the study period were included. No specific exclusion criteria were applied in order to maximize the inclusion of this rare clinical entity.

### 2.1. Study Variables

The following variables were analyzed:
Maternal characteristics: maternal age and obstetric history.Diagnostic characteristics: gestational age at the time of diagnosis.Prenatal ultrasound evaluation: morphological description of the main placenta and accessory lobe, precise determination of umbilical cord insertion site, color Doppler assessment of vascular connections between lobes, and transvaginal ultrasound to rule out vasa previa.Pregnancy outcomes: gestational age at delivery, mode of delivery (spontaneous vaginal, instrumental, or cesarean), and cesarean indications when applicable.Maternal and fetal complications: antepartum hemorrhage, placental abruption, retained placenta, postpartum hemorrhage, preterm labor (threatened preterm birth), and intrapartum fetal compromise.Neonatal outcomes: birth weight, Apgar scores at 1 and 5 min, and neonatal morbidity and mortality.


### 2.2. Ultrasound Evaluation Protocol

To assess the quality of the ultrasound diagnostic approach for succenturiate placenta, we developed a scoring checklist based on the completion of four key diagnostic steps:Complete morphological evaluation of the placenta.Accurate identification of umbilical cord insertion site.Color Doppler assessment to identify vascular connections between lobes.Transvaginal ultrasound to rule out vasa previa in selected cases.

Each properly documented step in the ultrasound report scored one point, while a lack of documentation scored zero. A maximum score of 4 was considered indicative of a complete and optimal examination.

### 2.3. Clinical Follow-Up

All patients with a diagnosis of succenturiate placenta underwent regular ultrasound monitoring following institutional protocols, with additional assessments based on individual clinical criteria. Obstetric management and mode of delivery were determined according to current clinical guidelines and taking into account patient-specific conditions patient-specific conditions. The clinical follow-up of the patients was conducted with a frequency adjusted to gestational age. In the early stages of pregnancy, biweekly evaluations were scheduled to ensure adequate monitoring of fetal and maternal development. As the pregnancy advanced and approached term, the frequency of visits was shortened to weekly controls, allowing for closer surveillance in the final weeks. This adjustment in periodicity was aimed at detecting any relevant changes and ensuring optimal care during the final stage of pregnancy.

### 2.4. Statistical Analysis

Given the small sample size, a descriptive analysis of the studied variables was performed. Quantitative variables were expressed as mean ± standard deviation or median and interquartile range, as appropriate. Qualitative variables were reported as absolute frequencies and percentages.

### 2.5. Ethical Considerations

This study was approved by the Institutional Research Ethics Committee. Data confidentiality and patient privacy were ensured in accordance with current regulations on personal data protection. This study was conducted in accordance with the Declaration of Helsinki and approved by the Ethics Committee of the CEI de Provincial Centre of Málaga, Spain, with protocol code FG-IMGP-2025 and date of approval date 20 April 2025.

## 3. Results

### 3.1. General Characteristics of the Study Population

A total of nine cases of succenturiate placenta were diagnosed and managed at the Department of Obstetrics and Gynecology of the Hospital Materno Infantil of Málaga during the study period (2014–2024).

### 3.2. Ultrasound Diagnosis

In most cases, the diagnosis was made during the routine second-trimester morphological scan (at 20–22 weeks of gestation), enabling appropriate monitoring throughout the rest of the pregnancy. Of the nine identified cases, five (55.6%) were diagnosed as isolated succenturiate placenta, while the remaining four cases (44.4%) were associated with velamentous insertion of the umbilical cord. No cases of vasa previa were recorded (Table 1).

### 3.3. Assessment of Diagnostic Quality

The quality of the ultrasound reports was evaluated using a four-point scoring system that reflected adherence to the key diagnostic steps for succenturiate placenta: complete placental morphology assessment, identification of the umbilical cord insertion site, color Doppler evaluation of communicating vessels, and transvaginal ultrasound to exclude vasa previa.

The results showed that the first three steps—placental morphological assessment, cord insertion determination, and color Doppler examination—were consistently performed in all cases. However, greater variability was observed in the use of complementary transvaginal ultrasound, which was the least frequently completed step. This variability may be attributed to the selective use of transvaginal scanning in cases where color Doppler raised suspicion of pathology near the cervical os (Table 2).

### 3.4. Obstetric Outcomes

The following results were obtained regarding the mode of delivery among the nine cases:Three cases (33.3%) resulted in spontaneous vaginal delivery.Two cases (22.2%) required instrumental delivery.Four cases (44.4%) were delivered via cesarean section.

Among the cesarean deliveries, three were elective and performed for the following reasons: occlusive placenta previa (one case), breech presentation (one case), and one case with no specific clinical indication documented. The remaining cesarean was performed on an emergency basis due to suspected fetal compromise (Table 2).

### 3.5. Maternal and Fetal Complications, Neonatal Outcomes

Maternal–fetal complications were observed in four out of the nine cases (44.4%):Intrapartum fetal compromise manifesting as bradycardia during the expulsive phase (two cases).Placental abruption (one case).Threatened preterm labor (one case)The following results were obtained with respect to neonatal outcomes:Eight out of the nine pregnancies (88.9%) reached full term (≥37 weeks of gestation).Birth weight was appropriate for gestational age in all cases.Apgar scores were favorable (≥7 at 5 min) in seven of the nine cases (77.8%). Complete Apgar data were unavailable for one case. One case of neonatal death (11.1%) was recorded and was associated with an instance of placental abruption.

## 4. Discussion

Succenturiate placenta represents an uncommon anatomical variant whose timely prenatal identification is essential to prevent potentially severe obstetric complications. This retrospective analysis of nine cases diagnosed at our institution over a 10-year period allows for a detailed examination of the clinical characteristics, management approaches, and perinatal outcomes associated with this condition, in comparison with findings from the current literature. We recognize that the detection of placental anomalies such as succenturiate lobes demands not only technical expertise but also a high index of diagnostic suspicion. In recent years, advances in high-resolution ultrasound and color Doppler have facilitated earlier and more accurate identification of placental variants. Nevertheless, the clinical impact of succenturiate placenta remains underreported and often underestimated in routine practice. The present discussion aims to contextualize our findings within the broader literature, offering insights into diagnostic trends, clinical implications, and future directions for standardizing care (Table 2).

### 4.1. Incidence and Diagnosis

The incidence of succenturiate placenta reported in the literature varies depending on population characteristics and diagnostic criteria. Estimates range from approximately 0.16% to 5% of pregnancies. This placental anomaly is associated with increased maternal age, assisted reproductive technologies, pelvic infections, and preeclampsia. Complications linked to succenturiate placenta include vasa previa, retained placental tissue, and postpartum hemorrhage [1]. In our series, the majority of diagnoses were made during the second-trimester anatomical scan—a critical window for detailed placental assessment—echoing the observations of Fadl et al. [2], who highlight this gestational period as optimal for anomaly screening.

A noteworthy finding was the relatively high prevalence of velamentous cord insertion (VCI), identified in 44.4% of cases, exceeding the rates reported in multicenter analyses such as that by Suzuki and Horio [3], which identified VCI in 25% of succenturiate placentas. This discrepancy may reflect enhanced diagnostic performance at our institution, likely facilitated by the use of a standardized and systematic ultrasound protocol for placental assessment.

### 4.2. Quality of Ultrasound Diagnosis

The diagnostic methodology implemented in our unit reflects strong adherence to best practices in placental sonography. In all cases (100%), we identified the morphological features of the main and accessory lobes, documented the site of cord insertion, and evaluated interlobar vascular connections via color Doppler. However, the use of complementary transvaginal ultrasound (TVUS) was inconsistently applied, typically reserved for cases with suspected low-lying lobes or vessels in close proximity to the internal cervical os (Figure 1).

This selective use of TVUS aligns with current ISUOG guidelines [11], which recommend transvaginal assessment in the presence of risk indicators for vasa previa or abnormal placental positioning. Moreover, our approach mirrors the diagnostic algorithm recently proposed by Kumar et al. [5], advocating for targeted rather than routine use of TVUS to balance diagnostic yield with resource allocation (Figure 2).

### 4.3. Obstetric Outcomes and Complications

The overall cesarean delivery rate in this group a patients (44.4%) was nearly twice the institutional average during the study period (~22%), supporting previous findings that link succenturiate placenta with an increased likelihood of cesarean section [6]. The most frequent indications were fetal malpresentation and placenta previa, although one case necessitated emergent surgical delivery due to acute fetal compromise.

Obstetric complications occurred in 44.4% of patients, with intrapartum fetal bradycardia being the most frequently reported event (22.2%). These findings resonate with a large systematic review by Stelzer et al. [7], who reported fetal compromise in 18.7% of similar cases. In a group a patients, one instance of placental abruption led to neonatal demise, underlining the potential severity of this variant, particularly when undiagnosed. Such outcomes are thought to result from the mechanical fragility of interlobar vessels, which may be particularly vulnerable during labor [8].

### 4.4. Postpartum Confirmation

In order to ensure diagnostic accuracy via ultrasound, particular emphasis was placed on the evaluation of placental remnants following delivery, which allowed for an appropriate clinical correlation. 

All cases were macroscopically confirmed at the time of delivery. In the case of two patients who delivered in non-affiliated centers, follow-up was conducted via telephone, and both provided medical reports documenting macroscopic confirmation.

However, not all cases underwent histopathological confirmation.

Of the nine cases diagnosed by ultrasound, the placenta was sent to the Department of Pathology in four cases, and confirmatory results were obtained in all of them.

### 4.5. Neonatal Outcomes

Despite the presence of placental anomalies, neonatal outcomes were largely favorable. Eight of the nine pregnancies reached term, with appropriate birth weights in all cases. Our rate of favorable Apgar scores (77.8%) is comparable to general obstetric populations, although one neonatal death associated with placental abruption highlights the importance of anticipatory management.

Interestingly, our data slightly differ from those of Chen et al. [9], who documented higher rates of prematurity and fetal growth restriction in placental anomaly cohorts. This discrepancy may be attributed to enhanced prenatal surveillance and individualized delivery planning in our unit.

### 4.6. Clinical Implications

These findings underscore the clinical relevance of incorporating systematic placental mapping into routine prenatal ultrasound evaluations. The structured four-step protocol implemented at our center—which includes morphological assessment, evaluation of cord insertion, Doppler vessel tracing, and selective transvaginal imaging—has demonstrated high effectiveness in characterizing placental morphology and informing obstetric management. This approach is consistent with the prior literature [11,12,13], which supports targeted surveillance to mitigate adverse perinatal outcomes.

It is noteworthy that the pregnancy complications observed were primarily limited to acute or emergency presentations. However, the most concerning and clinically significant complication in this series was placental abruption, which occurred in 11.1% of cases—a rate considerably higher than in the general obstetric population. This finding warrants particular attention, as placental abruption poses a substantial risk to both maternal and fetal health, often necessitating urgent intervention. The relatively elevated incidence observed suggests that pregnancies complicated by succenturiate placenta may have an underrecognized predisposition to abruption, especially in the absence of early identification and close surveillance.

Given this, it is essential to maintain a high index of suspicion in cases with confirmed or suspected succenturiate lobes, particularly when the accessory lobe is located near the uterine wall where vascular integrity may be more fragile. These data support the implementation of risk-based follow-up protocols and timely delivery planning to prevent maternal–fetal compromise.

In alignment with ISUOG guidelines [13,14], we advocate for the integration of individualized, risk-adapted algorithms that optimize the use of transvaginal sonography. Its application should be prioritized in cases where accessory lobes are detected in proximity to the internal cervical os or when Doppler interrogation reveals exposed or aberrant vessels that could predispose to vasa previa or hemorrhagic complications [15,16].

### 4.7. Study Limitations

We acknowledge several limitations to this study, including its retrospective design, small sample size, and lack of a control group. Additionally, as a single-center analysis, our results may not be generalizable to settings with different resource levels or diagnostic protocols. Nevertheless, the consistency of our diagnostic criteria and clinical pathways offers a robust platform for proposing best practice recommendations.

### 4.8. Proposed Diagnostic and Clinical Management Algorithm

Based on and supported by international evidence, we propose the following algorithm for the diagnosis and management of suspected succenturiate placenta:1.**Routine second-trimester morphology scan (20–22 weeks)**Systematic evaluation of placental shape and continuity.Assess for separate echogenic masses suggestive of accessory lobes.2.**Color Doppler mapping**Identify vessels traversing fetal membranes between lobes.Evaluate the umbilical cord insertion site (central, marginal, velamentous).3.**Selective transvaginal ultrasound**Indicated when accessory lobes are low-lying or Doppler imaging reveals vessels near the internal cervical os.4.**Follow-up and delivery planning**Intensified monitoring if velamentous insertion or interlobar vessels are present.Consider elective cesarean delivery at 34–36 weeks if vasa previa is confirmed.Ensure active management during the third stage of labor to prevent retained placental tissue.

This structured and pragmatic approach can aid in minimizing perinatal risks associated with succenturiate placenta and should be considered for inclusion in institutional prenatal care protocols (Figure 3).

## 5. Conclusions

Succenturiate placenta requires detailed ultrasonographic evaluation and tailored obstetric monitoring. Our findings confirm the association with velamentous cord insertion and the increased risk of maternal–fetal complications described in the literature. The diagnostic approach implemented at our institution demonstrates a high degree of adherence to international recommendations; however, there remains room for standardizing the use of complementary transvaginal ultrasound in selected cases.

Future prospective studies with larger sample sizes and the application of standardized diagnostic protocols may contribute to optimizing prenatal management and reducing the incidence of complications associated with this condition.

## Figures and Tables

**Figure 1 diagnostics-15-01329-f001:**
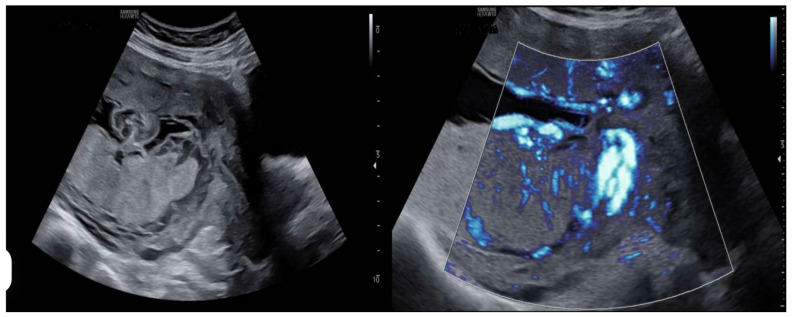
Two-dimensional ultrasound study of a case diagnosed with succenturiate placenta. A main placental mass is observed on the posterior wall, along with a smaller accessory lobe on the anterior wall. In the second image, a color Doppler study is performed to identify communicating vessels between the two lobes and determine their location.

**Figure 2 diagnostics-15-01329-f002:**
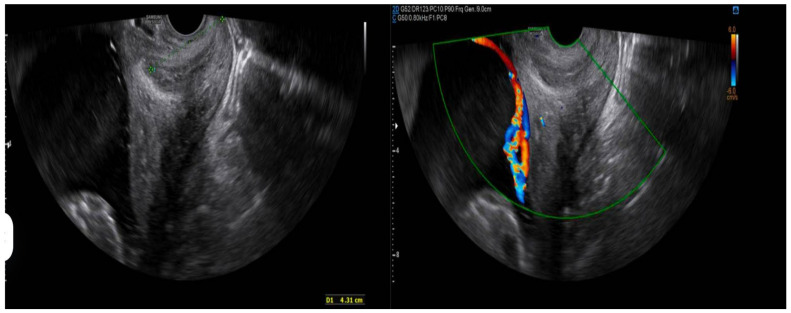
Transvaginal ultrasound study performed to rule out an association with vasa previa in one of the study patients, revealing a highly suggestive image on grayscale imaging. As shown in the second image, the addition of color Doppler was particularly elucidative, demonstrating high-flow vessels crossing the internal cervical os (ICO).

**Figure 3 diagnostics-15-01329-f003:**
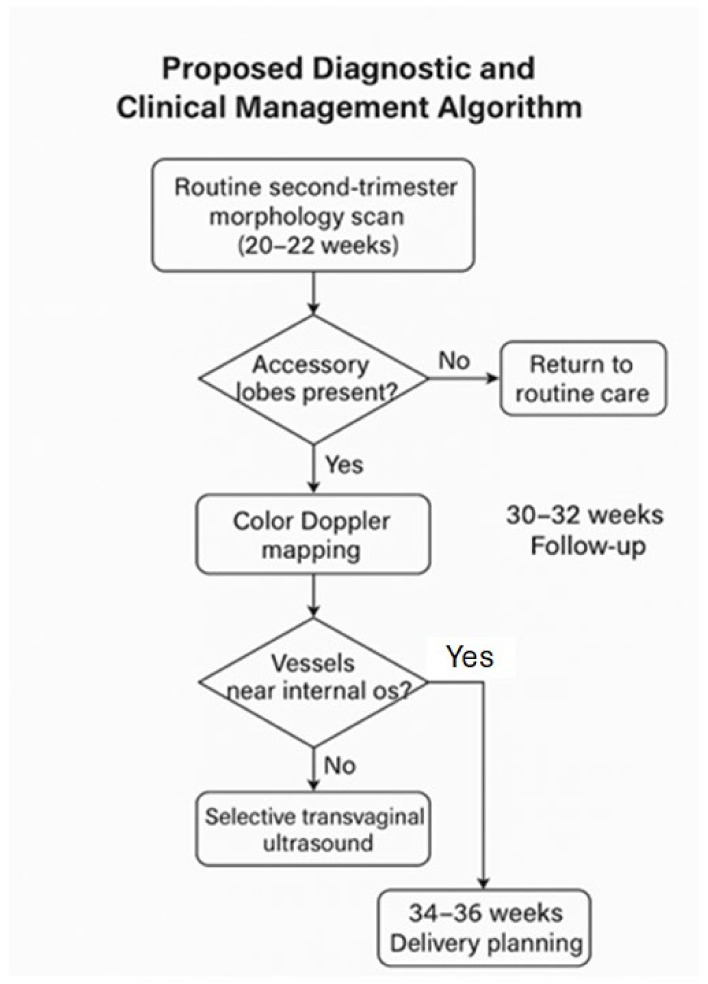
Clinical Management Algorithm.

**Table 1 diagnostics-15-01329-t001:** Main data of interest about pregnancy, diagnosis, delivery, and outcomes of all 9 cases included in the study. (VI = velamentous insertion).

Year	GA (wks)	Anomaly	Labor Onset	Delivery Type	Complications	Weight	Apgar
2023	21 + 3	Succenturiate placenta + IV	Cesarean (breech)	Central extrafundal cesarean	-	3280	9/10
2023	38 + 2	Succenturiate placenta	Cesarean (breech)	Cesarean	-	2930	9/10
2022	41 + 3	Succenturiate placenta	Induction due to EVP	Vaginal (eutocic)	-	3180	9/10
2017	34 + 4	Succenturiate placenta	Spontaneous	Vaginal (eutocic)	IUGR 34w	2340	9/10
2012	39 + 1	Succenturiate placenta	Spontaneous	Extrafundal vaginal delivery	-	2890	9/10
2011	39 + 1	Succenturiate placenta	Spontaneous	Instrumental (PFBF)	Expulsive bradycardia	2640	9/10
2005	39 + 3	Succenturiate placenta + IV	Spontaneous	Emergency cesarean for PROM + S	-	2670	9/10
2013	42 + 2	Succenturiate placenta + IV	Spontaneous	Instrumental (PFBF)	Expulsive bradycardia	2960	9/10
2007	37 + 6	Succenturiate placenta + IV	Spontaneous	Emergency cesarean	Abruption (PDNIP)	2650	0/1

**Table 2 diagnostics-15-01329-t002:** Diagnostic characteristics and maternal–fetal outcomes in cases of succenturiate placenta.

Characteristic	n (%) or Mean ± SD
** Diagnosis **	
Isolated succenturiate placenta	5 (55.6%)
Succenturiate placenta + velamentous cord insertion	4 (44.4%)
Succenturiate placenta + vasa previa	0 (0%)
** Pregnancy Outcome **	
Spontaneous vaginal delivery	3 (33.3%)
Instrumental delivery	2 (22.2%)
Cesarean section	4 (44.4%)
** Maternal–Fetal Complications **	
Intrapartum fetal bradycardia	2 (22.2%)
Placental abruption (DPPNI)	1 (11.1%)
Threatened preterm labor (TPL)	1 (11.1%)
No complications	5 (55.6%)
** Neonatal Outcomes **	
Term birth (≥37 weeks)	8 (88.9%)
Favorable Apgar score (≥7 at 5 min)	7 (77.8%)
Appropriate birth weight for gestational age	9 (100%)
Neonatal mortality	1 (11.1%)

Abbreviations: SD: standard deviation; DPPNI: premature detachment of a normally inserted placenta; TPL: threatened preterm labor.

## Data Availability

The original contributions presented in the study are included in the article; further inquiries can be directed to the corresponding author.

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
