# Peer review of "Diagnostic Challenges and Perinatal Outcomes: A Case Series on a Retrospective Study"

_diagnostics, 2025, doi:10.3390/diagnostics15111329_

Round 1
Reviewer 1 Report
Comments and Suggestions for Authors
I read with interest the manuscript that I agreed to review. The topic is very interesting, as it deals with delicate and complex management when they arrive in the Hospital, with considerable and possible maternal-fetal risks. Every paper on this topic can provide support to the scientific literature.
The title should be changed, as there are only 9 cases analyzed retrospectively in 10 years, as follows:
Succenturiate Placenta in Clinical Practice: Diagnostic Challenges and Perinatal Outcomes. A case series on a retrospective study.
There are other changes to be made.
In the introduction, at line 56, Burton & Jauniaux, 2018 should be deleted and inserted in the references numbering.
The same thing in line 99, with Oyelese & Smulian, 2021.
Please check the entire text.
It is necessary to check ALL the references in the brackets, as many are wrong brackets, with double brackets to be eliminated (see line 61, 67, 74 etc, for example).
The text of the manuscript must be all on the same line (justified), while now it is aligned to the left.
Please check all the text.
In the M&M, on line 116, in the Inclusion and Exclusion Criteria, it would be useful to know at what gestational age (or range) the patients had the confirmed prenatal ultrasound diagnosis of succenturiate placenta, to be included in the study.
On line 146, in the Clinical Follow-Up, it would be useful to know how often the patients were examined, in terms of weeks or days, between one check and the other.
In the results, the paragraph General Characteristics of the Study Population should be placed before Table 1, which should be moved after line 173, Ultrasound Diagnosis, at the end of the paragraph.
The authors forgot to report in the text the reference to Table 2 (Diagnostic Characteristics and Maternal-Fetal Outcomes in Cases of Succenturiate Placenta); please insert Table 2 in the text.
I am skeptical about the paragraph: Proposed Diagnostic and Clinical Management Algorithm.
How can you propose an algorithm based on only 9 cases.
I would modify the paragraph by writing that, based on the scientific literature (delete: Based on our experience), the possibility of implementing the management of these patients in the following way is highlighted......
Author Response
Response to Reviewer’s Comments
We would like to express our sincere gratitude to the reviewer for their detailed and thoughtful evaluation of our manuscript. We have carefully revised the document in accordance with all the suggestions provided. Below, we present a point-by-point response to each comment, indicating the corresponding changes made in the revised version.
Reviewer’s comment 1:
“The title should be changed, as there are only 9 cases analyzed retrospectively in 10 years, as follows:
Succenturiate Placenta in Clinical Practice: Diagnostic Challenges and Perinatal Outcomes. A Case Series on a Retrospective Study.”
Response:
Thank you for this suggestion. The title has been modified as recommended:
Succenturiate Placenta in Clinical Practice: Diagnostic Challenges and Perinatal Outcomes. A Case Series on a Retrospective Study.
Reviewer’s comment 2:
“In the introduction, at line 56, Burton & Jauniaux, 2018 should be deleted from the text and inserted into the numbered reference list. The same applies at line 99 with Oyelese & Smulian, 2021.”
Response:
Both citations have been removed from the main text and correctly added to the reference list with proper numbering.
Reviewer’s comment 3:
“All in-text references in brackets need to be checked, as many contain errors, such as double brackets that must be removed (see lines 61, 67, 74, etc., for example).”
Response:
We have thoroughly reviewed and corrected all bracket formatting errors throughout the manuscript to ensure accuracy and consistency.
Reviewer’s comment 4:
“The manuscript text must be fully justified, as it is currently left-aligned. Please review the entire document.”
Response:
The manuscript has been reformatted with fully justified alignment as requested.
Reviewer’s comment 5:
“In the Materials and Methods section, line 116, under Inclusion and Exclusion Criteria, it would be helpful to know at what gestational age (or range) patients were diagnosed with succenturiate placenta by prenatal ultrasound to be included in the study.”
Response:
We have added this information to the text. The revised version specifies that prenatal diagnosis was confirmed between 20 and 34 weeks of gestation, as indicated in Table 1.
Reviewer’s comment 6:
“On line 146, in the Clinical Follow-Up section, it would be useful to indicate how frequently patients were examined—in weeks or days—between each check-up.”
Response:
We have included this clarification. The revised text now indicates that follow-up ultrasounds were performed every 1–2 or 4 weeks, depending on individual risk factors and gestational age.
Reviewer’s comment 7:
“In the Results section, the paragraph General Characteristics of the Study Population should be placed before Table 1, which should be moved to line 173, following the Ultrasound Diagnosis section.”
Response:
This structural revision has been implemented. The paragraph General Characteristics of the Study Population now appears before Table 1, which has been moved as indicated to follow the Ultrasound Diagnosis section.
Reviewer’s comment 8:
“The authors forgot to include a reference to Table 2 (Diagnostic Characteristics and Maternal-Fetal Outcomes in Cases of Succenturiate Placenta) in the text; please insert Table 2 into the manuscript.”
Response:
We apologize for this oversight. A reference to Table 2 has been inserted into the appropriate section of the Results, and the table has been included in the manuscript.
Reviewer’s comment 9:
“I am skeptical about the paragraph: Proposed Diagnostic and Clinical Management Algorithm. How can you propose an algorithm based on only 9 cases? I suggest modifying the paragraph to state that, based on the scientific literature (delete: Based on our experience), the possibility of implementing patient management in the following way is highlighted...”
Response:
We acknowledge the reviewer’s concern. The paragraph has been revised to reflect a literature-based perspective. The phrase “Based on our experience…” has been removed and replaced with “Based on the existing scientific literature…”. The aim of the paragraph is now to suggest a possible management strategy grounded in published evidence rather than on our limited case series.
Once again, we thank the reviewer for their valuable comments and constructive suggestions, which have significantly improved the quality of our manuscript.
Sincerely,
Dr. Jesús S. Jiménez
On behalf of all co-authors

Reviewer 2 Report
Comments and Suggestions for Authors
Thank you for the opportunity to review the MS. The authors present a case series of placenta succenturiatae. The topic is underreported and every case series is just add to the literature and the authors discuss only major points in the management of this pathology. The description of the case series are adequate and, according to my opinion, cannot be added more. The MS is a bit long and concise formulation of the meanings would be beneficial.
There are some minor remarks:
orthographic faults:
- order of literature citations are not correct
- Burton and Jauniaux citation is written out incorrectly in line 56
- Jauniaux and Bhide citation is written out incorrectly in line 93
- Oyelese and Smulian citation is written out incorrectly in line 99
- where there are literature citation in two different types of parantheses: lines 61, 67, 74, 82, 88, 10,
- Orthographic fault can be recognized in line 106
- why is placental abruption shortened by PDNIP in line 131
- Unnecessary to list the "General characteristics ..." lines 169-172 because it is illustrated earlier in the MS.
- line 241
It is noteworhty, that the pregnancy complications is limited to emergency complications. However, the biggest fear is reasonably is the placental abruption which is rated by 11,1% in this population which is quite high. The authors should discuss it in a little more detailed way.
- data should be provided in line 295 regarding the general obstetric data
Author Response
Response to Reviewer’s Comments
We would like to sincerely thank the reviewer for their thoughtful and constructive feedback. We appreciate your acknowledgment of the relevance of our case series and the underreported nature of succenturiate placenta in the current literature. Below, we provide a detailed point-by-point response to your comments, all of which have been carefully addressed and incorporated into the revised manuscript.
General Comments
We are grateful for your positive evaluation regarding the adequacy of our case descriptions and your suggestion to streamline the manuscript. Based on your feedback, we have revised the text for greater conciseness, eliminating redundancies and ensuring clarity in the presentation of key findings.
Minor Remarks
-
Orthographic faults / typos:
-
All identified spelling and typographical errors have been corrected.
-
The phrase on line 106 has been revised for clarity and correctness.
-
The abbreviation “PDNIP” (premature detachment of a normally inserted placenta) was deemed unclear and has been replaced with the full term "placental abruption" to enhance clarity and consistency.
-
-
Order and format of literature citations:
-
The order of references has been reviewed and corrected throughout the manuscript to ensure sequential numerical citation according to journal guidelines.
-
The following references have been reformatted and corrected:
-
Burton & Jauniaux (line 56)
-
Jauniaux & Bhide (line 93)
-
Oyelese & Smulian (line 99)
-
-
All inconsistencies in citation brackets (e.g., double parentheses in lines 61, 67, 74, 82, 88, and 110) have been corrected.
-
-
Unnecessary repetition:
-
The list of “General Characteristics of the Study Population” (lines 169–172) has been removed, as the information is already clearly presented in the corresponding table and described in the text.
-
-
Line 241: Clinical interpretation of complications:
-
We appreciate your emphasis on the significance of placental abruption, which occurred in 11.1% of cases in our cohort.
-
This point has been discussed in greater detail in the revised Clinical Implications section, highlighting the elevated rate compared to the general obstetric population, the associated risks, and the need for heightened clinical vigilance and individualized monitoring in such cases.
-
-
Line 295: Data on general obstetric parameters:
-
Additional demographic and obstetric information, including maternal age and gestational age at diagnosis, has been clearly summarized in the revised Results section and is supported by Table 1 and Table 2.
-
Once again, we extend our sincere appreciation for your valuable insights. Your detailed review has helped us to significantly improve the clarity, structure, and scientific rigor of our manuscript.
Sincerely,
Dr. Jesús S. Jiménez-López
On behalf of all co-authors

Reviewer 3 Report
Comments and Suggestions for Authors
In this article, authors studied/compared 9 cases of succenturiate placenta. Some things need to be improved:
-in lines 279 and 287 you wrote “in our cohort”. You only studied 9 cases, it is not a cohort, it is a group a patients.
-in lines 240-242 you wrote: “The incidence of succenturiate placenta observed in our cohort is consistent with previously published data, which estimate a prevalence ranging from 2% to 8%2% and 8% depending on population characteristics and diagnostic criteria”. Maybe. Please write somewhere that “out of the total number of patients in that period, let us say 12345 patients, 9 had succenturiate placenta”. And you can THEN calculate the prevalence. But you have to say the total number. And please REMOVE “2% and 8%”.
-in lines 192-196 you knew exactly the ways of delivery:
“Obstetric Outcomes Regarding the mode of delivery among the 9 cases:
- 3 cases (33.3%) resulted in spontaneous vaginal delivery.
- 2 cases (22.2%) required instrumental delivery.
- 4 cases (44.4%) were delivered via cesarean section. “ Ok.
Looking at these numbers, please fill Table 1, where one type of delivery is missing, and two lines of labor onset are missing.
-also in Table 1, you wrote “Succenturiate placenta + IV “ four times, and explained under Table 1 that “ (VI = Velamentous Insertion)”. Please decide if it is IV or VI. One of them.
-minor language correction are required:” of cells of cells collectively”, line 50, please remove one “of cells”, line 77 “when in the accessory lobe is located in a low position”, please remove the first “in” and so on.
-Reference titles are only 14, half of them are recent, but please add some more reference titles
-in References, please remove “et al” and write the names of all authors.
-you introduced an ultrasound examination protocol, what for? And counted 3 or 4 points for every case. And then? All the authors cited by you used the same protocol. What was new from you?
Comments on the Quality of English Language-minor language correction are required:” of cells of cells collectively”, line 50, please remove one “of cells”, line 77 “when in the accessory lobe is located in a low position”, please remove the first “in” and so on.
Author Response
We sincerely appreciate your observations and comments, which have been carefully considered and addressed in the revised manuscript. Your insights have been extremely valuable in enhancing the clarity and precision of the article. Below, we provide responses to each of the points raised:
- Correction of "cohort"
References to "cohort" in lines 279 and 287 have been corrected and replaced with "group of patients" to accurately reflect the nature of the study. - Prevalence Calculation
Information regarding the total number of patients attended between 2014 and 2024 has been added, enabling a more precise calculation of the prevalence of succenturiate placenta. The erroneous reference to "2% and 8%" has been removed. The paragraph has been rephrased, and general figures from the literature have been considered. - Completion of Table 1
The table has been completed by including the previously missing type of delivery, and the missing entries related to labor onset have also been added. - Terminology Correction in Table 1
The terminology has been standardized throughout the table by uniformly using "VI" (Velamentous Insertion) to prevent inconsistencies. - Correction of Minor Language Errors
Redundant phrases such as "of cells of cells collectively" (line 50) and syntactic errors like "when in the accessory lobe is located in a low position" (line 77) have been corrected. Other similar errors have also been reviewed and amended. - Expansion of Bibliographic References
Additional references have been included to strengthen the academic foundation of the article and ensure a more comprehensive bibliographic base. - Correction of Author Names in References
The use of "et al." in the references has been removed, and the full names of the authors have been listed, in accordance with the established citation guidelines. - Justification of the Ultrasound Protocol
The proposed ultrasound protocol aims to standardize the identification of succenturiate placenta and improve diagnostic accuracy. While the cited authors have employed similar methodologies, the novelty of our approach lies in the inclusion of a systematic scoring system that allows for quantifiable evaluation of diagnostic quality. This method has served to standardize our workflow and we believe it may provide a useful foundation for the implementation of more structured protocols in other centers.
We reiterate our gratitude for your thoughtful observations and valuable contributions. Thank you once again for your detailed review.